# Progress on the Haystack Observatory Postprocessing System

**Daniel Hoak** *,†, **John Barrett** †, **Geoffrey Crew** †  and **Violet Pfeiffer** †

Massachusetts Institute of Technology, Haystack Observatory, 99 Millstone Rd, Westford, MA 01886, USA
* Correspondence: dhoak@mit.edu
† These authors contributed equally to this work.

**Abstract:** The Haystack Observatory Postprocessing System (HOPS) is a multipurpose tool for post-correlation calibration and data analysis in Very-Long Baseline Interferometry experiments. The requirements on stations, baselines, and bandwidth for the Next Generation Event Horizon Telescope (ngEHT) have motivated a significant refactoring of the HOPS codebase. In this paper, we present the requirements, specifications, and design of HOPS 4.0 and the current state of the refactoring, and we discuss future work.

**Keywords:** VLBI; black holes; signal processing

## 1. Introduction

In Very-Long Baseline Interferometry (VLBI), the signal from widely separated radio observatories is correlated between each pair of antennas, known as a *baseline*, to generate complex time-averaged quantities, known as *visibilities*. A critical step between correlation and further data analysis (e.g., imaging) is solving for corrections to the relative difference in arrival times of the wavefront at the antennas, which can be caused by geometric path-length differences, atmospheric effects, and instrumental effects that are not accounted for in the timing model used for correlation. The procedure that solves for these residual delay and delay-rate solutions that maximize the visibility amplitude on each baseline is known as *fringe fitting*.

The Haystack Observatory Postprocessing System, or HOPS, is a multipurpose software package designed to facilitate fringe fitting, phase calibration/correction, and data analysis for VLBI experiments. HOPS has a multi-decade history as a VLBI tool, beginning with work by Alan Rogers in FORTRAN in the 1970s, followed by a complete rewrite into C by Colin Lonsdale and Roger Cappallo in the 1990s. There have been incremental improvements since then by a large cast of contributors. The primary tool in HOPS is the fringe-fitting program `fourfit`; separate tools provide data summary and visualization methods (`aedit`, `alist`), and there are functions to segment, merge, average, export, and incoherently search data (`fringex`, `fourmer`, `average`, `CorAsc2`, and `search`, respectively).

The Event Horizon Telescope (EHT) Collaboration has recently used VLBI techniques to image the horizon scale structure of two supermassive black holes, M87* [1] and Sagittarius A* [2]. The HOPS software was a critical component of the EHT data-processing pipeline [3,4], along with CASA [5–7] and AIPS [8]. The next generation EHT (ngEHT) is currently being designed and is expected to dramatically expand the EHT network [9]. The ngEHT is planned to include up to 30 stations recording four, 2-bit, dual-frequency, dual-polarization channels at 8 GHz sampling frequency. These design goals amount to a 10x increase in the number of baselines and a 4x increase in the total bandwidth compared to the EHT, which exceeds the capabilities of the current HOPS software[1]. Thus, as part of the ngEHT design effort, the HOPS software is being significantly refactored to support the expansion of the network.

## 2. Goals of the Refactoring

The current HOPS software (major version 3, or HOPS3[2]) is written in C and dates from the 1990s. While HOPS3 has had tremendous success as a tool for the VLBI community, the design of the existing codebase has several limitations. We will address these by refactoring the existing functions and methods into C/C++ for HOPS4.

The memory allocation of the pipeline is controlled by hard-coded parameters that place a limit on the number of stations, channels/sub-bands/IFs[3], accumulation periods (APs), and other dimensions of the data. HOPS4 will use dynamic memory allocation and will have no practical limit on the number of stations, baselines, channels, and APs in a fringe search.

Currently, HOPS has the ability to perform phase corrections on a per-channel basis, but it cannot perform fully complex corrections (amplitude and phase) at the sub-channel level. HOPS4 will support amplitude, phase, and delay corrections at each frequency bin.

The existing code relies on legacy software packages that are no longer supported, such as the plotting utility PGPLOT [10], which is deeply integrated into the HOPS code. HOPS4 will decouple plotting and analysis routines and provide new plotting tools using modern packages such as `matplotlib` [11]. HOPS4 will also provide hooks for user-defined plotting packages.

HOPS requires a user-generated configuration file to set basic analysis parameters. The syntax for this file is complex and does not support operations such as flagging or vetoing data beyond a rudimentary manual selection in time or sub-band (channel). Furthermore, the code base as a whole is monolithic and difficult to extend or use in a modular way for either debugging or analysis. HOPS4 will use modern wrappers (e.g., Python) for initializing the configuration parameters and refactor the code into modular, independent libraries.

Most importantly, HOPS4 will be capable of performing any operation that HOPS3 is capable of and will continue to support current data formats.

The following sections describe particular design choices for HOPS4.

### 2.1. Data Format

HOPS processes VLBI data that have been correlated using the DiFX software correlator [12,13]. DiFX output (which contains the complex visibilities) is in the so-called "Swinburne" format. Currently, HOPS requires users to convert the DiFX output files into the legacy "Mark4" format using the `difx2mark4` utility. The Mark4 data format is based on I/O methods from the tape-drive era, and the format of the in-memory data structures in HOPS3 is tightly coupled to this disk storage format. As a result, modifications to the in-memory structures require similar modifications to the disk storage format and vice versa. HOPS4 has replaced the Mark4 disk storage format with a binary data format that allows increased flexibility and improved file I/O performance. A new utility, `difx2hops`, converts the visibilities and metadata from the Swinburne format into the HOPS4 format.

The binary file format is well-suited for large homogeneous data files such as visibilities. These can be several gigabytes in size for a single (few minute) EHT scan. Heterogeneous data types, such as experiment metadata, are currently stored in memory as C structures with hard-coded sizes. HOPS4 stores these data types as JSON key-value pairs. The JSON format supports lists and is not restricted by compile-time size definitions.

Finally, the HOPS4 team has implemented support for the `vex` [14] and `ovex` metadata formats, including the new `Vex 2.0` [15] format. The metadata from these files are stored as key-value pairs in a JSON object. HOPS4 will support output of fringe data to other formats such as `uvfits` or `hdf5`.

## *2.2. Data Structures*

Currently, HOPS imports data from the Mark4 data format into C-type structures, whose parameters are hard-coded and are difficult to modify. HOPS4 will utilize C++ template classes to construct multidimensional arrays of any trivially constructable data type. This feature provides a method for augmenting the code with a wide variety of possible data types that all share the same unified array-like access interface. The base class has methods to handle memory allocation, resizing, data access via indices or iterators, and data operators for streaming, and so new data types can be defined with minimal effort to meet new use-cases without requiring extensive changes to the code base. While the dimensionality and element types composing a data array in HOPS4 must be fixed at compile time (for example, channelized visibility data with four dimensions: time, frequency, channel, and polarization), the size of each dimension is not fixed or limited. This allows HOPS4 to analyze datasets of arbitrary size. In addition, these template classes support labeling each axis with intervals defined by key-value pairs to facilitate data flagging.

The data operations that act upon multidimensional data in HOPS4 follow a common interface that only requires methods to set the data objects that are used in the operation, initialize the internal state, and execute the operation. This allows users to chain operations in an ordered list for execution, while allowing each operation to be self-contained and unit-testable.

Furthermore, we are implementing bindings and plugins to expose the data structures to external methods. We are using the `pybind11` and `SWIG` libraries to construct bindings to the channelized visibility containers, which allows users to manipulate the data with Python code. We also verify that the operator interface works with OpenCL extensions, which allows the most time-consuming data operations to be parallelized on GPUs.

## *2.3. Plotting and Data Summary*

HOPS currently generates plots using the PGPLOT graphics library, which has been unsupported for over a decade and has become difficult to install on contemporary operating systems. Unfortunately, the PGPLOT functions are deeply embedded in the HOPS code. Significant refactoring is required to separate the data processing from plotting and file output.

The HOPS4 analysis methods will be completely independent of plotting and visualization tools. The analysis code will export results to standard file formats, and the plotting functions will read the data from the disk. Our default plotting routines will be written in `matplotlib`, but the modularity of the plotting functions will enable users to implement their own plotting methods (and allow HOPS to be compiled and executed without linking to any plotting libraries).

The basic plotting result from HOPS is the *fringe plot* (see Figure 1), which has been widely recognized by VLBI users for many years. The fringe plot will be replicated in HOPS4, but options will be available for different use-cases, for example replacing the cross-power spectrum with an alternative plot, or simplifying the dense text and metadata at the bottom of the figure.

The HOPS `alist` utility summarizes the results of a VLBI experiment in an "A-file", a text file that records the useful parameters for each scan, baseline, and polarization. The utility `aedit` provides a number of command-line tools that manipulate A-list data, such as plotting, filtering, editing and sorting. Currently, `aedit` includes both command-line and GUI interfaces. The GUI capability currently requires PGPLOT, which will be replaced in HOPS4. We have prototyped a PyQt-based GUI that replicates the graphical interface of `aedit` and supports the command-line features.

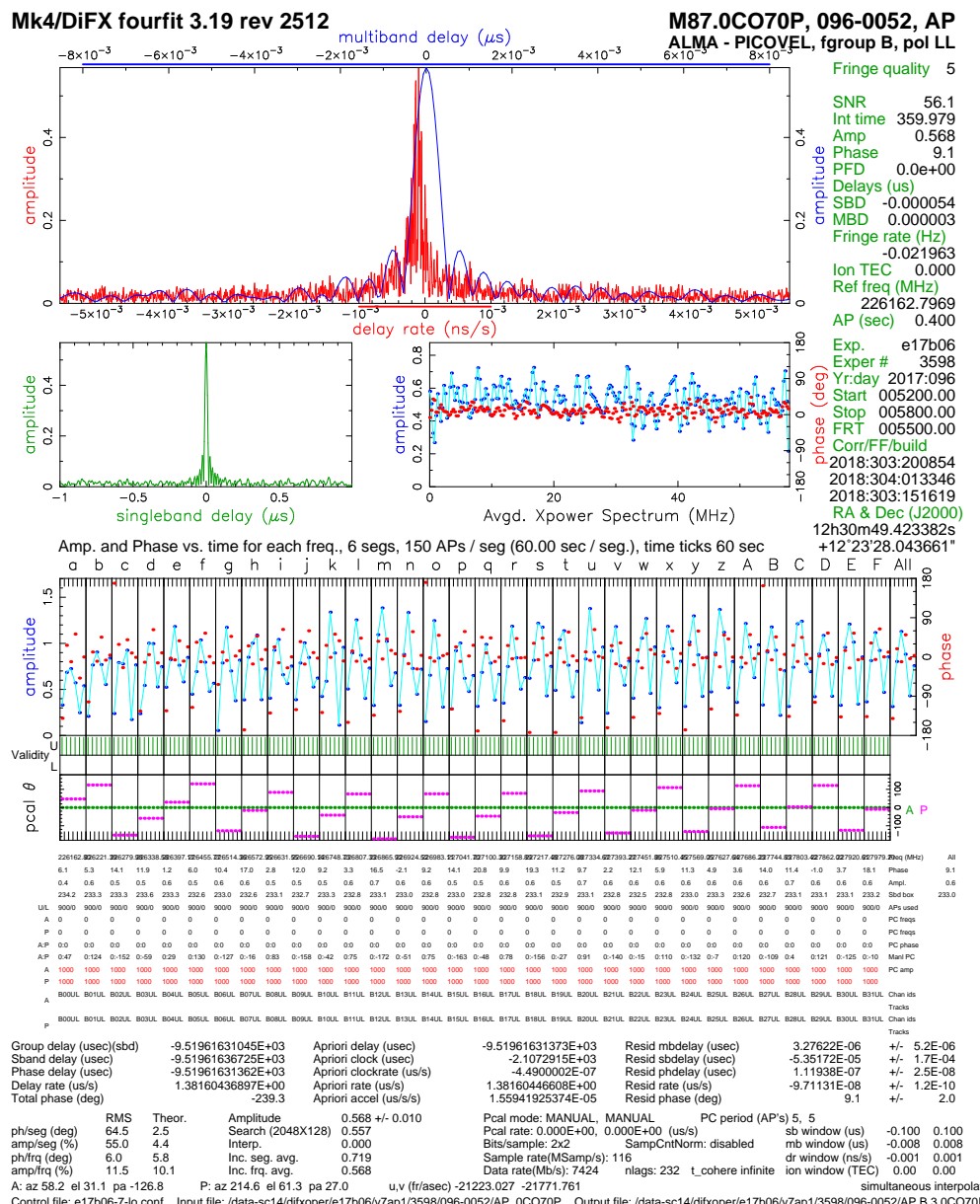

**Figure 1.** An example fringe plot from the 2017 EHT observations of M87.

## 3. Testing

The HOPS4 team is following best practices with regards to test-driven development. These include regression, unit, component, and integration tests. We plan to use HOPS3 as a test oracle and currently support test-coverage reports. The suite of tests is intended to provide developers with up-to-date verification of the required HOPS functionality across the supported platforms and distributions. It also provides end users with build-time checks to indicate that the installation was successful. We use coverage tools to demonstrate that the test plan executes a satisfactory fraction of the code, and that all required functions and cases are exercised by the tests. Additionally, we collect performance assessments and benchmarks on a regular basis using captured datasets.

## 4. Future Work

HOPS currently fringe-fits VLBI data on a per-polarization, per-baseline basis; each pair of polarizations and stations are treated independently. Closure quantities are not considered during the fringe-finding stage, so non-closing errors can occasionally be intro-

duced with this method unless care is taken to iteratively generate a global fringe solution by repeated re-fringing. A number of global fringe-fitting algorithms exist (e.g., [16,17]) that minimize the residual phase, delay, and delay-rate solutions for arrays with three or more stations and can aid in recovering fringes on baselines with low signal-to-noise. One of the goals of the refactoring effort of HOPS is to modify the software so it may accommodate a choice of alternative algorithms for fringe fitting in addition to the native baseline-based algorithm.

Sources with weak continuum emission but bright spectral lines can be imaged using spectral-line VLBI techniques. In principle, the current version of HOPS can perform spectral-line VLBI, but the procedure is quite technical and requires a significant amount of hand-tuning. Implementing a fringe-fitting algorithm that supports spectral-line VLBI by searching for the fringe maximum over delay-rate and frequency space is a highly desirable goal for HOPS4.

HOPS3 calculates per-baseline, per-scan fringe solutions in a one-shot execution of `fourfit` from the command line; crude parallelization can be made by running multiple `fourfit` jobs over multiple datasets, but this must be orchestrated by the user. While this sort of simple parallelization is crucial for processing large swaths of independent data needed for VLBI-imaging, it is not particularly useful for acceleration on data with granularity below that of a single baseline/scan. The basic fringe-fitting algorithm in HOPS3 computes the Fast-Fourier Transform (FFT) and searches for the fringe maximum in the three-dimensional space defined by the single-band delay, the multi-band delay, and the delay rate; the computationally intensive portion of this method can be parallelized relatively easily either with multi-threading or via single-instruction–multiple-data (SIMD) techniques. Given the raw computational power of modern GPUs, the SIMD avenue is particularly attractive as it is well-tailored for the repeated calculation of simple mathematical functions (e.g., the delay/delay-rate phase rotation) as well as array manipulation, which dominates the fringe-fitting computation. OpenCL extensions of simple data operations (array scaling/multiplication) have been demonstrated and will be applied to additional operations as computational bottlenecks in the existing and future algorithms are identified.

### 5. Conclusions

The refactoring of the HOPS VLBI analysis software for the ngEHT is well underway. Approximately 23k lines of code have been written and define new data structures and I/O routines, import/export to legacy data formats, and perform data analysis on the new structures. The goal of the refactoring is to maintain the current functionality and performance of HOPS while supporting the increased number of stations, baselines, and frequency bands for the ngEHT. HOPS4 will rely on standard software packages (C/C++, Python, and minimal associated tools) that are readily available on common Linux-based operating systems such as Ubuntu, Debian, and CentOS. HOPS4 will have improved modularity and extensibility compared to HOPS3, allowing users to export data to common formats, inject code to test new methods, and ease debugging.

The HOPS development team[4] welcomes user feedback, questions, and feature requests. The team looks forward to releasing a beta version of `fourfit` in early 2023 and a beta version of the full HOPS4 software package in 2024.

**Author Contributions:** Authors D.H., J.B., G.C. and V.P. contributed equally to this work. All authors have read and agreed to the published version of the manuscript.

**Funding:** This research was supported by the National Science Foundation through grant numbers AST-1935980 and AST-2034306.

**Data Availability Statement:** Not applicable.

**Acknowledgments:** The authors gratefully acknowledge useful discussions from their MIT Haystack, EHT, and ngEHT colleagues.

**Conflicts of Interest:** The authors declare no conflict of interest.

## Notes

[1] HOPS3 has hard-coded limits on the number of stations (16), baselines (120), channels/sub-bands (64), and accumulation periods (8192) that are challenging to modify in the current architecture.

[2] The latest release as of this article is version 3.24: ftp://gemini.haystack.mit.edu/pub/hops (accessed on 15 December 2022)

[3] In VLBI, each spectral band measured at the antenna is typically divided into several smaller bands for averaging and analysis. In HOPS, these sub-bands are called *channels*, while in other fringe-fitting packages (e.g., AIPS) they are called intermediate-frequency bands or "IFs". For example, the EHT collects data in 2 GHz bands, which in HOPS are divided into 32 channels, each 58 MHz wide; see Figure 3 in [4]. One or more individual frequency bins ("channels" in AIPS) are referred to as a "sub-channel" in HOPS.

[4] hops-dev@mit.edu

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
