# Peer review of "Progress on the Haystack Observatory Postprocessing System"

_galaxies, doi:10.3390/galaxies10060119_

Round 1

Reviewer 1 Report

Thanks the efforts of the Haystack Observatory Postprocessing System (HOPS) development team, the updating data structures in HOPS and the parallelized processing in fourfit look quite attractive to VLBI users. I have no major comments on this manuscript as a progress report of technical component related to the data processing for the next generation Event Horizon Telescope (ngEHT). Here there are some minor comments..

1.     Line26-29, please elaborate this part e.g. the features and requirements of ngEHT on HOPS, current description is more like a general progress report rather than a dedicated paper to the special issue of ngEHT.

2.     Line 39-41, the function here is for the bandpass calibration in VLBI data by my understanding, my question is where can we obtain the gain, phase, and delay corrections at each frequency bin through HOPS? As fourfit output is already channel averaged, it is not possible to directly measure intra-channel phase bandpass from detected fringes (Blackburn et al. 2019)

3.     The functions proposed in section 4 is promising. I wonder if the similar ability to pre-correct nonlinear phase evolution over time using station-based ad hoc phases, but to pre-correct nonlinear delay evolution over time is possible? This may benefit (sub-)mm VLBI to calibrate the atmospheric delays and phases among multi-frequency bands. In other words, please consider to highlight some functions related to future sub-mm VLBI.

Author Response

We thank the reviewer for their helpful comments. We have made the following changes to the manuscript:

1. We have added some text that describes the expansion of stations and bandwidth required by the ngEHT and how it relates to the current version of HOPS.

2. The current version of HOPS does not support gain corrections, and as you point out only applies per-channel phase corrections (and, time-varying phase corrections can only be applied in a laborious way per-scan). HOPS4 will support intra-channel gain and phase corrections, and make any calculated intra-channel amplitude/phase corrections available to the user.

3. This is a good idea, and one we intend to support in HOPS4. Currently, applying these sorts of corrections is not possible, but HOPS4 users will have the ability to multiply each channel by a complex-valued, time-varying function. 

Reviewer 2 Report

This short paper provides an overview of the current status and near-term plans for the next generation of HOPS software.  The paper is clearly written and to the point, and I believe that it merits publication in this special issue.  I have provided below a few minor comments that the authors should address, but I do not view any of them as a barrier to publication.

===========================================

Comments:

Lines 12-13: In the spirit of continuing the helpfully pedantic approach that this paper's intro is already taking (e.g., by briefly defining what a "baseline" is and what "visibilities" are), it would make sense to additionally provide a brief description of what a "delay" and a "delay rate" are here (and perhaps even what is meant by "residual" in this context).

Lines 17-18: Are there citations that go along with the Rogers and Lonsdale+Cappallo versions of HOPS?  This seems like an appropriate place to reference any original publications or memos.

Line 27: It would make sense to provide a reference here to point the reader to what exactly is meant when it is said that the ngEHT will "dramatically expand the EHT network."

Footnote 1: I am able to access the Haystack server specified in this footnote, but is there (or is it planned to have) a more public-facing repository (e.g., Github) for the HOPS software?  This would facilitate development of the codebase by users outside of the immediate Haystack community.

Lines 39-40: What is the distinction between a "channel" and a "sub channel"?  I'm not familiar with the latter.

Line 100: The word "demonstrated" feels odd here without either a reference or a presentation of the demonstration in this paper.  Perhaps "verified" would be more appropriate?

Line 147: The word "molecular" here seems unnecessarily specific (there do exist examples of non-molecular spectral lines, even in VLBI data); probably "spectral" is more appropriate.

Author Response

We thank the reviewer for their helpful comments. We have made the following changes to the manuscript:

Lines 12-13: We have added a few words to this explanation, which hopefully thread the needle between VLBI pedagogy and brevity.

Lines 17-18: Unfortunately there is no publication that describes the initial version of HOPS.

Line 27: This has been expanded.

Footnote 1 (now footnote 2): Yes, we expect to make a public git repository available for HOPS4.

Lines 39-40: We have added some text to explain this more completely: a HOPS "channel" is a sub-band (same as an AIPS IF), and a HOPS sub-channel is one or more frequency bins (or AIPS "channels").

Line 100: Fixed, thank you.

Line 147: Fixed, thank you.

Reviewer 3 Report

This is a useful progress report on HOPS4, which will have commendable and significant improvements over HOPS3. See below for some minor comments.

Line 9-13: Some basic terms are explained here, others are not. One could mention the Fourier relationship between visibilities and the image, and briefly explain what delays and delay rates are, how they originate in the observations, and why we need to calibrate them out.

Line 15: Maybe briefly explain what ‘calibration’ entails in HOPS beyond fringe fitting (e.g. does it do amplitude calibration).

Line 16-18: Is it possible to provide references to this early work?

Line 26-27: Please cite Doeleman et al. (2019).

Line 35-38: Could you give an estimate of current limits, and by how much they will be exceeded by the ngEHT? E.g. how much could the EHT expand before memory allocation becomes a problem? Apart from the ngEHT, do you envision other arrays or observations for which HOPS4 could be particularly useful?

Line 87-88: What is the difference between frequency and channel here? Should uvw-coordinates and/or stations be in this list?

Line 181: Is it possible to give a rough timeline for the release? Should I think of a month, 6 months, years?

Author Response

We thank the reviewer for their helpful comments. We have made the following changes to the manuscript:

Line 9-13: We have expanded this section, and tried to thread the needle between pedagogy and brevity.

Line 15: Modified to explicitly say that HOPS performs phase corrections. In section 2 we explain that HOPS3 cannot perform amplitude corrections (but HOPS4 will).

Line 16-18: Unfortunately there is no publication that describes the initial version of HOPS.

Line 26-27: We are not sure which publication the reviewer is referring to; is it arXiv:1909.01411, the white paper for Astro2020? We have added this citation.

Line 35-38: We have added some descriptions of the limitations in HOPS3. The issue is not memory allocation, but hard-coded limits in the structures that are difficult to change without complicated downstream effects. It's quite a headache; HOPS4 will have no limits on these parameters.

HOPS is a critical tool for geodetic data analysis with the VGOS network, and we expect HOPS4 will expand on this legacy.

Line 87-88: The data array described here represents the visibility data imported from the correlator, and the axes of this array are polarization (L or R), sub-band or channel (1-32), frequency bin within the channel, and time. We have tried to explain this more clearly. The uvw coordinates, stations, and other metadata are not stored in arrays, since their format is heterogeneous. HOPS3 stores these in C-structures with hard-coded formats; HOPS4 will store them as key-value pairs.

Line 181: We plan to release a beta version of fourfit on the timescale of months, and release a beta version of the full software package in approximately one year. We've added this information to the conclusion.

Reviewer 4 Report

This is a concise and well written description about the ongoing development of the multi-purpose tool for post correlation and data analysis of VLBI data, called HOPS. In particular, some details for the major transition and updates from HOPS3 to HOPS4 are presented.

My main comment is about the lack of a clear timeline for the development and release of the new HOPS4 pipeline. It is mentioned that it will be released in the near future in the conclusions, but it would be desirable to have a bit more quantitative estimate and put it in the context of the ngEHT timeline.

Other than this, all the sections are brief, clear and well organized, and I have no major comments.

Two minor typographical comments:

- line 111: us -> use

- line 159:define the FFT acronym

Author Response

We thank the reviewer for their helpful comments. We have fixed the typographical errors in the latest manuscript.

For the timeline, we have updated the conclusion to say that we expect to release a beta version of fourfit on the timescale of months, and release a beta version of the full software package in approximately one year.